# Formulation, Preparation, Characterization, and Evaluation of Dicarboxylic Ionic Liquid Donepezil Transdermal Patches

**DOI:** 10.3390/pharmaceutics14010205

**Published:** 2022-01-16

**Authors:** Linh Dinh, Soohun Lee, Sharif Md Abuzar, Heejun Park, Sung-Joo Hwang

**Affiliations:** 1College of Pharmacy, Yonsei University, 85 Songdogwahak-ro, Yeonsu-gu, Incheon 21983, Korea; dinhkhanhlinh@yonsei.ac.kr (L.D.); soooo3937@naver.com (S.L.); sumonzar@gmail.com (S.M.A.); 2Yonsei Institute of Pharmaceutical Sciences, Yonsei University, 85 Songdogwahak-ro, Yeonsu-gu, Incheon 21983, Korea; 3College of Pharmacy, Duksung Women’s University, 33 Samyangro 144-gil, Dobong-gu, Seoul 01369, Korea

**Keywords:** donepezil, Alzheimer’s disease, dicarboxylic acid, ionic liquid, transdermal drug delivery, transdermal patch

## Abstract

Donepezil (DPZ) is generally administered orally to treat Alzheimer’s disease (AD). However, oral administration can cause gastrointestinal side effects. Therefore, to enhance compliance, a new way to deliver DPZ from transdermal patch was developed. Ionic bonds were created by dissolving dicarboxylic acid and DPZ in ethanol, resulting in a stable ionic liquid (IL) state. The synthesized ILs were characterized by differential scanning calorimetry, optical microscope, Fourier transform infrared spectroscopy and nuclear magnetic resonance spectroscopy. The DPZ ILs were then transformed to a suitable drug-in-adhesive patch for transdermal delivery of DPZ. The novel DPZ ILs patch inhibits crystallization of the IL, indicating coherent design. Moreover, DPZ ILs and DPZ IL patch formulations performed excellent skin permeability compared to that of the DPZ free-base patch in both in vitro and ex vivo skin permeability studies.

## 1. Introduction

Alzheimer’s disease (AD), which has affected over 40 million elderly people, is the most common cause of dementia and one of the leading causes of death in vulnerable patients [1]. However, the mechanism underlying the occurrence of AD is not fully understood, and there is currently no cure for this disease, thus patients with AD become a considerable burden on their families and society [1,2]. The most prescribed agents that can temporarily manage dementia are cholinesterase inhibitors (ChEIs) such as donepezil, galantamine, and rivastigmine [3]. Among these ChEIs, donepezil (DPZ) has been reported to show beneficial effects in patients with more severe AD [4]. DPZ is generally administered orally in the treatment of AD [5]. However, oral administration can cause gastrointestinal side effects such as diarrhea, nausea, and vomiting due to drastic changes in plasma concentration upon oral administration [5]. In particular, most patients with AD refuse to take drugs or have difficulty in swallowing or chewing drugs due to dysphagia. Since the oral DPZ therapy is associated with adverse events and plasma concentration fluctuations, DPZ transdermal patch has been employed as an alternative route of administration. Traditional DPZ transdermal preparations contain many permeation enhancers owing to its low skin permeability, which may cause skin irritation problems; moreover, solid crystals are generated during storage, resulting in reduce adhesion and a non-uniform skin permeation rate. In this study, we developed a DPZ ionic liquid (IL) transdermal patch—a novel transdermal preparation of DPZ that can overcome the current drawbacks.

DPZ is an acetylcholinesterase inhibitor used to treat cognitive symptoms in mild to moderate AD [3,4,5]. In particular, it appears to benefit both cognitive ability and mental function [4,5]. DPZ, 2-[(1-benzyl-4-piperidyl) methyl]-5,6-dimethoxy-2,3-dihydroinden-1-one, is a piperidine-based agent that is chemically unique among all ChEIs. DPZ has been reported to be superior to others because of its high potency and selectivity for acetylcholinesterase in the central nervous system [5]. DPZ reversibly inhibits acetylcholinesterase and alleviates the symptoms of AD by increasing the concentration of acetylcholine and improving choline function. However, due to its low solubility (<0.1 mg/mL (insoluble) in H_2_O), DPZ still has limited applications as a potential therapeutic agent. Being a small molecular weight (179.31 g/mol (free base) and lipophilic (log *p* value of 3.08–4.11), DPZ has been considered to be a well-suited active pharmaceutical ingredient for transdermal delivery.

Transdermal drug delivery systems (TDDS) have the advantages of being non-invasive, self-administered, and having long-term drug action thus reducing dosing frequency. TDDS helps improve both patients’ and caregivers’ compliance compared to the parenteral and oral administration, especially in old patients. Moreover, TDDS provides a stable, uniform drug blood levels resulting in reduced side effects. Patches also permit rapid termination of drug delivery by stripping in case of any severe side effect experiences or signs of an overdose.

Several studies have shown ILs as promising active pharmaceutical ingredients in the development of transdermal formulation [6,7,8,9]. ILs have large scale industrial applications in this field over the past few years. ILs can bypass the barrier properties of the outermost layer of the epidermis and diffuse intracellularly through mechanisms such as disruption and fluidization of cellular integrity, generation diffusion pathways and extraction of lipid components of the stratum corneum [10,11]. An IL is defined as an organic salt in its liquid state [12]. The stable liquid phase is maintained over a wide range of low to high temperature [13]. ILs can be classified into different categories based on their thermodynamic and physicochemical properties: e.g., high-temperature molten salt, network-forming IL, low-temperature molten salt, and room temperature IL. In this study, we focused on room temperature ILs with melting point below 100 °C but not necessarily above 0 °C. Room temperature ILs typically contain large organic cations with basic cyclic structure and/or quaternized nitrogen and/or phosphorus atoms with attached alkyl chains.

Similar to most other active pharmaceutical ingredients, DPZ exists in a solid crystalline form. The most obvious problem with crystalline form is polymorphism. Polymorphism can negatively affect solubility, and ultimately the absorption and bioavailability of the drug [9]. In this study, DPZ was solubilized with several counterions (dicarboxylic acid) in ethanol (the solvent) and eventually converted to ILs. The aim of this study was to create a room temperature IL of DPZ (DPZ solubilizing form) that can suppress crystal formation for a prolonged duration to facilitate the sustained release of DPZ from a transdermal patch.

## 2. Materials and Methods

### 2.1. Materials

DPZ free-base was purchased from Perrigo (Allegan, MI, USA). Adipic acid, glutaric acid, isophthalic acid, itaconic acid, phthalic acid, saccharic acid, sebacic acid, terephthalic acid, and α-ketoglutaric acid were purchased from Deajung Chemicals and Metals (Siheung, Korea). Fumaric acid was purchased from Sigma Aldrich (St. Louis, MO, USA). Azelaic acid and pimelic acid were purchased from Alfa Aesar (Haverhill, MA, USA). Maleic acid and malic acid were purchased from Junsei Chemicals (Tokyo, Japan). Succinic acid was purchased from Samchun Pure Chemicals (Gyeonggi, Korea).

Ethanol and polyethylene glycol (PEG) 400 was purchased from Samchun Chemicals (Gyeonggi, Korea). Tween 80 was purchased from TCI Chemicals (Tokyo, Japan).

Water was purified using Milli-Q^®^ Reference water purification system (Merck Millipore, Alsace, France).

DURO-TAK^®^ 87-2051, DURO-TAK^®^ 87-2074, and DURO-TAK^®^ 87-2196 were provided by Henkel (Dusseldorf, Germany).

Chloroform-D (D, 99.8%) containing 0.05% *v*/*v* tetramethylsilane was obtained from Cambridge Isotope Laboratories (Andover, MA, USA).

Female SKH1 hairless mice for animal studies were purchased from YoungBio (Seongnam, Korea). The mice were housed in standard cages placed in a semi-specific pathogen-free facility at 19 ± 1 °C and 50 ± 5% relative humidity, under a 12-h light–dark cycle. The mice were allowed to access to food and water freely prior to the experiments. All experiments were approved by the Institutional Animal Care and Use Committee (IACUC) at Yonsei University, Seoul, Korea, and were performed according to the IACUC guidelines.

### 2.2. Methods

#### 2.2.1. Donepezil (DPZ) Ionic Liquid (IL) Preliminary Study

A number of dicarboxylic acids (co-former candidates) were selected for the preliminary study. The synthesis of ILs consists in an acid-base reaction. The base, in this case DPZ free base, was mixed with the acid, respectively). The solvent evaporation method was used for the preparation of DPZ ILs between DPZ and an appropriate co-former. First, 50 mg each of DPZ and co-former (1:1, *w*/*w*) were accurately weighted, then gently dissolved in 1 mL of ethanol. The mixture was then put in a dry oven (Lab companion, Daejeon, Korea) at 60 °C overnight to evaporate the solvent and impurities (residual non reacted and water). After the evaporation process, visual observation and characterization analysis of the samples was performed. The obtained ILs presented as a limpid and viscous liquid.

#### 2.2.2. Preparation of DPZ ILs

In this study, ILs were prepared using the solvent evaporation method. First, 50 mg of DPZ and co-former (1:1 weight ratio) were gently dissolved in 1 mL of ethanol. The solvent then was evaporated at 60 °C overnight in a dry oven. The DPZ ILs thus obtained were stored at 25 °C in 2 mL tubes sealed with paraffin films to avoid moisture.

#### 2.2.3. Differential Scanning Calorimetry (DSC)

The thermal behavior of the samples was identified by DSC technique using a DSC Auto Q2000 (TA instrument, New Castle, DE, USA). All samples were accurately weighed to 3–5 mg in an aluminum pan. Powder samples (DPZ free base) were sealed in aluminum pans with lids. IL samples were sealed in aluminum pans with hermetic lids. Samples were scanned from −60 °C to 200 °C at a heating rate of 10 °C/min under dry nitrogen at a constant flow rate of 40 mL/min. An empty pan was used as reference. For IL samples, the tightness of the pan was tested by weighing to avoid evaporation (if any).

#### 2.2.4. Optical Microscopy

Optical microscopy allowed a magnified view of the DPZ IL surface. This technique helps to confirm the amorphous form of the DPZ IL. Images were taken at five times magnification using a microscope camera (Axio Cam MRc, Carl Zeiss, Oberkochen, Germany). Luminous intensity was controlled using a light source (Hal 100, Carl Zeiss, Oberkochen, Germany).

#### 2.2.5. Fourier Transform Infrared Spectroscopy (FT-IR)

Infrared spectra of the samples were recorded using Cary 630 FT-IR spectrometer (Agilent Technologies, Santa Clara, CA, USA) equipped with an attenuated total reflectance (ZnSe crystal) to confirm the interaction between DPZ and co-former. Each spectrum was scanned in the range of 400–4000 cm^−1^ with a resolution of 8 cm^−1^, and was derived from single average scans collected in the mid-infrared region (2.5 to 50 μm) at a high spectral resolution; a total of 32 scans were obtained.

#### 2.2.6. Nuclear Magnetic Resonance Spectroscopy (NMR)

The 1H NMR spectra of all IL samples in deuterated Chloroform-D (CDCl_3_) were recorded at room temperature using JNM-ECZ600R 600MHz spectrometer (Jeol, Tokyo, Japan). Residual 1H resonance from deuterated solvent (1H of CDCl_3_ at 7.25 ppm) was used as the reference 1H spectrum with the methyl resonance of tetramethylsilane at 0.0 ppm, according to the IUPAC recommended method. Taken in consideration that the chloroform can react with the amines [14], NMR were taken as soon as possible to minimize possible reaction.

#### 2.2.7. Equilibrium Solubility Determination

An excess amount of DPZ was added to 5 mL of water or buffer in 10 mL vials. DPZ ILs were synthesized in 10 mL vials, then 5 mL water or buffer was added in the vials. The vials were then placed in a water bath shaker at 25 °C, and 100 rpm for 24 h. The remaining solid was removed by filtration. All samples were filtered using a 0.2 µm polytetrafluorethylene syringe filter before analysis, and DPZ concentration was determined by high-performance liquid chromatography (HPLC).

#### 2.2.8. In Vitro Skin Permeation Test

In vitro DPZ IL permeation test was performed using Franz diffusion cell (Transdermal diffusion cell drive console with tilt, Logan Instruments, Somerset, NJ, USA) with an orifice size of 15 mm. The Franz diffusion cell system consists of a 2 mL donor compartment and a 12 mL receptor compartment. Each diffusion cell was maintained at a constant temperature (32 ± 0.5 °C), under continuous stirring at 300 rpm using a magnetic stirrer. Strat-M™ (Merck Millipore, Burlington, MA, USA) artificial skin was used for this experiment. Standard phosphate buffered saline (PBS) was prepared, and 0.5 M sodium hydroxide was used to adjust the pH to 7.4, which served as the acceptor solution and the donor solution. The artificial skin was soaked for 10 min in PBS solution before testing for hydration. Then, an equal amount of DPZ and DPZ IL were casted on the artificial skin. Furthermore, 0.2 mL of DPZ suspension at a concentration of 14 mg/mL and used as the reference. At predetermined time points, 1 mL of solution was taken from the receiver compartment and immediately replaced with 1 mL of the fresh preheated medium. During the permeation test, the sink condition in the receptor compartment of the Franz diffusion cell was maintained such that the drug concentration did not exceed 30% of the maximum solubility of the receptor solution. Air bubbles underneath the membrane were removed via the side arm. The concentration of each sample was determined by HPLC as described below. The cumulative permeation amount (*Q*, μg/cm^2^) was calculated using the following equation, where *A* is the effective diffusion area (cm^2^), *C* is the concentration of the donepezil in the receiver phase (μg/mL), *i* and *j* are samples numbers, *V* is the volume of the receiver chamber (mL) and *v* is the volume of the collected sample (mL).
(1)$$Q=\frac{C_{i}\times V\times \sum_{j=1}^{i-1}C_{j}\times v\;}{A}$$

#### 2.2.9. Acrylic Adhesive and DPZ IL Compatibility Test

The acrylic adhesive and DPZ IL compatibility tests were carried out to confirm that the DPZ IL and acrylic pressure sensitive adhesive can be mixed uniformly to form a transparent solution. The acrylic pressure sensitive adhesive was selected among: DURO-TAK^®^ 87-2074, DURO-TAK^®^ 87-2196, and DURO-TAK^®^ 87-2051 manufactured by Henkel (Dusseldorf, Germany). Ten milligrams each of active pharmaceutical ingredient (DPZ), co-former (1:1 *w*/*w*), and 80 mg of acrylic adhesive were accurately weighed, mixed together, and stored in an air-tight glass vial. The combination that produces a clear patch solution was selected.

#### 2.2.10. Drug-In-Adhesive Patch Preparation

Pre-determined amounts of active pharmaceutical ingredient (DPZ), co-former and acrylic adhesive were accurately weighed and put together in an air-tight glass vial. The mixture was then dissolved in 1 mL of ethanol and was stirred for 1 h using magnetic stirrer until a homogenous mixture was formed. The DPZ IL patch was dissolved in ethanol and stored at room temperature for 12 h until casting. Casting was applied on the release liner with a wet film thickness of 100 μm, and the casting was done using a Knife coating device (Kipae E&t, Gyeonggi, Korea) at a speed of 5.0 mm/s. Casted patches were dried at 60 °C for 30 h in a dry oven. After the drying process, a backing layer was attached to the upper side of the patch.

#### 2.2.11. DPZ IL Patch Formulation Development

Polyethylene glycol (PEG) 400 and Tween 80 were used respectively as plasticizers and surfactants to improve the permeability of the DPZ IL patch. Drug-in-adhesive patch solution was prepared as mentioned above with the addition of PEG 400 and Tween 80 at an appropriate amount.

#### 2.2.12. Ex Vivo Permeability Test

Several DPZ IL patch formulations were selected as model IL patches after comprehensively considering the physicochemical and biochemical properties, compatibly test results, and permeability factors. Ex vivo skin permeability experiments were performed to evaluate the permeability of these prepared patch samples. Ex-vivo skin permeability experiments were performed by using 6-week-old, female, SKH1 hairless mice. The hairless mice were sacrificed by CO_2_ gas exposure immediately before the experiments. The abdominal and dorsal skin was cut and clamped between the donor and receptor cells of the Franz diffusion cell with the stratum corneum side facing the donor cell. Ex vivo skin permeability experiments used Franz diffusion cells similar to in vitro skin permeability experiments. The temperature of the Franz diffusion cell was maintained at 37 ± 0.5 °C. Then, the prepared patch samples were applied on the hairless mouse skin. At predetermined time points, 1 mL of solution was taken from the receiver cell and immediately replaced with 1 mL of the fresh medium.

#### 2.2.13. HPLC Analysis

DPZ IL and transdermal patch samples were analyzed using 1200 Infinity series (Agilent technology, Santa Clara, CA, USA). The HPLC system consisted of a pump (1260 Quat Pump VL), column heater (1260 TCC), auto sampler (1260 ALS), and UV detector (1260 VWD VL). DPZ was separated by an ODS-C18 column, (4.6 mm × 250 mm, particle size of 5 μm) (Phenomenex, Torrance, CA, USA) and column temperature was set to 25℃. The mobile phase consisted of 50% 0.1 M phosphate buffered solution pH 2.7, 20% acetonitrile, and 30% methanol. The flow rate was set to 1 mL/min. All solvents used for analysis were of HPLC grade. The injection volume was 20 μL and the UV detection wavelength was 268 nm. A stock solution of DZP was prepared up to a concentration of 100 μg/mL by serial dilution with the mobile phase. The lower limit of quantification was approximately 100 ng/mL and the lower limit of detection was approximately 50 ng/mL.

#### 2.2.14. Statistical Analysis

All the measurement data are expressed as the mean (*n* = 3).

Statistical analyses were carried out using the IBM SPSS Statistics (Statistical Product and Service Solutions (SPSS) software (version 25.0, IBM, New York, NY, USA). The ANOVA and two-sided Bonferroni tests were applied to analyze the differences in skin permeability between the artificial skin and hairless mouse skin groups. Statistical significance was set at *p* < 0.05.

## 3. Results and Discussion

### 3.1. Donepezil Ionic Liquid (DPZ IL) Preliminary Study

In the preliminary study, visual checks of the samples were done to confirm the formation of ILs using the selected co-formers and DPZ. Table 1 presents a list of the co-former candidates.

All the co-formers are dicarboxylic acids and are commercially available; were selected based on the Generally Recognized as Safe (GRAS) database approved by the U.S. Food and Drug Administration (FDA) for transdermal products.

The ingredients were dissolved in ethanol, and then the solvent was completely evaporated to obtain the final ILs. DPZ free base reacted with the acid and the reaction products are a salt of DPZ-coformer complex, the protic ionic liquid itself [15]. A total of 13 DPZ ILs (DPZ-adipic acid, DPZ-azelaic acid, DPZ-glutaric acid, DPZ-itaconic acid, DPZ-maleic acid, DPZ-malic acid, DPZ-phthalic acid, DPZ-pimelic acid, DPZ-sebacic acid, DPZ-succinic acid, DPZ-suberic acid, DPZ-tartaric acid, and DPZ -α ketoglutaric acid) were synthesized, and it was visually observed that they changed from solid or semi-solid to viscous liquid at room temperature. In the case of not forming ILs, the solid form was maintained at room temperature. This can be explained by the complexity of the ions forming ILs, the formation of ILs may require more than the number cations and anions present in the prepared mixture and a certain degree of ionization of acid or base ions needed should be calculated. So far, a common trait which is used to determine an IL is its “melting point”, nevertheless, the scarce knowledge of the built-in ions of ILs is still a barrier to completely characterize an IL [12]. Aspartic acid, fumaric acid, glutamic acid, isophthalic acid, saccharic acid and terephthalic acid did not form ILs with DPZ, respectively. The appearances of the formed ILs are summarized in Figure 1.

### 3.2. Characterization of DPZ ILs

DSC analysis was performed to confirm that the formed DPZ ILs behaved in a liquid state (gel/paste-like state) at room temperature. The glass transition temperatures (Tg) of the DPZ ILs were observed in the range of −29.36 °C to −11 °C, as detected by DSC. The DSC analysis results confirm that the prepared DPZ ILs exist in a gel/paste form with fluidity at room temperature. Figure 2 shows the DSC thermograms of DPZ ILs and Tg of each IL formulation. The Tg values of DPZ-adipic acid, DPZ-azelaic acid, DPZ-glutaric acid, DPZ-itaconic acid, DPZ-maleic acid, DPZ-malic acid, DPZ-phthalic acid, DPZ-pimelic acid, DPZ-sebacic acid, DPZ-succinic acid, DPZ-suberic acid, DPZ-tartaric acid and DPZ-α ketoglutaric acid were −21.14 °C, −22.99 °C, −18.49 °C, −27.41 °C, −20.57 °C, −21.39 °C, −11 °C, −29.36 °C, −15.19 °C, −21.74 °C, −17.52 °C, −13.83 °C, and −13.10 °C, respectively.

Thermal analysis confirmed the glass transition at very low temperatures, and the transition of a solid phase into a liquid phase occurred very soon due to the formation of IL, indicating a stable liquid state aligned with a thermal stability up to 60 °C. The results herein are in agreement with previous study showing that ILs with a lower Tg showed better fluidity [16]. Tg could be decreased through modification by the anionic component [16], and the anions of the dicarboxylic acids may be responsible for the intermolecular interaction, thus affecting the cohesive energy of the ILs, resulting in low viscosity and low Tg.

Optical microscopy was performed to visually confirm the liquid phase of the ILs consisting of DPZ and the co-former. Figure 3 shows DPZ, a white powder at room temperature. Figure 4 shows the optical microscopic images of DPZ ILs.

Figure 5 shows the FT-IR spectra of DPZ and DPZ ILs, which are quite similar except some peak shifts, indicating the absence of chemical reaction during IL preparation. We first focus on the region between 3600 and 2700 cm^−1^, which is dominated by single bonds to hydrogen interactions. Because carboxylic acids usually exist as hydrogen-bonded dimers, they show a very broad band for the O–H stretch. This is also the same region of the sharp C–H stretching bands of both alkyl and aromatic groups. The expansion of the O–H stretch and the less intense, shifted C–H in ILs could be explained by the hydrogen bonds formed via N–H in DPZ and the aromatic groups of the acids. The interactive forces depend on the alkyl chain length, which result in a decreased electrostatic attraction between the cation and the anion. The original sharp peak of C=O at 1684 cm^−1^, C=C at 1590 cm^−1^, and C–N at 1307 cm^−^^1^ in the structure of DPZ were shifted, and peak broadening occurred in the ILs results, explaining the bending mode of the absorbed H_2_O and aromatic ring stretch, but they were maintained in the spectrum of the IL formulation. Vibrational bands in the fingerprint region 1500–500 cm ^−1^ characterize single bond interactions, including cation–anion interactions. The coordination between the anion and DPZ was further confirmed and calculated using NMR analysis.

A molar ratio of 1:1 between DPZ and conformer was calculated by 1H NMR for all ILs. The detailed NMR spectrum of each IL can be found in Appendix A section.

DPZ peaks remained almost the same in the 13 NMR spectra of the ILs. First, we noticed at the signals of neighboring hydrogen attached to the piperidine and benzene ring of DPZ. There was no obvious difference in the chemical shift of the benzene ring of DPZ between the DPZ ILs and DPZ. The two peaks at approximately δ 3.83 ppm were originally from the two -OCH3 of DPZ, which shifted toward the right end when DPZ was incorporated with the co-former. This could be explained possibly by the formation of hydrogen bonds. The appearance of water was also seen in NMR spectra of ILs, water stayed in the IL system, connected to DPZ cation and to the carboxylic anion at the other end (cation–water–anion interaction) [17]. In addition, the chemical shift at approximately δ 3.66 ppm, δ 2.41 ppm, and/or δ 2.51 ppm indicated that the nitrogen at the piperidone of DPZ was protonated in the presence of the conformer, resulting in a chemical shift in that region, and the number of H atoms which donated to DPZ from the co-former is also shown (Appendix A).

The ILs described herein were synthesized based on the reaction between the two anions of the dicarboxylic acid group and the dual cation-π inhibitors of acetylcholinesterase. Taking advantage of the two ionization states of di-carboxylic acids (co-former) terminated by carboxylic acids, which are derived from the anhydrides, DPZ reacted to the co-former to form a stable room temperature ILs.

Both FTIR and NMR spectra showed distinct properties of DPZ. Based on these findings, we have now convinced that our synthesized DPZ ILs is approaching to the concept of deep eutectic solvents—binary mixtures of compounds. Deep eutectic solvents are a class of ionic liquids that is highly thermally stable, low volatility and a novel promising class of green material. Here, DPZ ILs are combination of two components: acid and base or so called anionic and cationic species. The two components formed a “eutectic liquid” at lower temperature than melting temperature of either of the individual component. In addition, natural carboxylic acids are considered renewable sources. They can be easily prepared in larger scale. Therefore, our DPZ ILs’ biodegradability is high and their toxicity is expected to be low.

The solubility of DPZ and DPZ ILs in water and 0.01 M PBS (pH 7.4) are listed in Table 2. DPZ free base exhibited very poor water solubility (0.017 mg/mL) and the solubility of DPZ increased slightly to 0.018 mg/mL in PBS pH 7.4. The solubility of DPZ ILs in water was higher than that of DPZ and increased when placed in PBS buffer solution. When DPZ—malic acid IL and DPZ—alpha ketoglutaric IL were dissolved in water and PBS buffer, significant high concentrations of DPZ were detected, indicating more than 100-fold increase of solubility in water and PBS buffer. All ILs showed higher level of solubility than that of DPZ free base in water and PBS buffer. This could be explained by the hydrogen-bond donor–acceptor interaction of the DPZ multicomponent systems. In case of ILs containing active pharmaceutical ingredients, the contribution of counterions to total molar mass should be taken in consideration therefore mass per volume unit can be converted to mol per volume unit for better comparison. DPZ free base exist in crystalline state and cannot form hydrogen bonds, but the appearance of salt-forming acids and water facilitated the protonation of nitrogen atoms of the piperidine of DPZ.

### 3.3. In Vitro Skin Permeation Tests

In vitro skin permeation test was performed to compare the skin permeability of DPZ and DPZ ILs. The amount of drug permeated (μg/cm^2^) from the applied dose was calculated using Equation (1) for DPZ free base powder and DPZ ILs is shown in Figure 6. In vitro skin permeation tests results showed that the skin permeability of DPZ ILs (390–1475.93 μg/cm^2^) was significantly higher than that of the DPZ free base suspension (111 μg/cm^2^) during 24 h. In particular, DPZ-glutaric acid ILs showed to have the highest skin permeability followed by that of DPZ-adipic acid ILs. In agreement with Hao Wu et al., DPZ ILs were more permeable than DPZ free base in terms of transdermal flux [8]. All DPZ ILs showed good skin permeation profile without reaching to the saturation state, DPZ free base (used as a control) showed quite low permeation profile. At the same concentration, DPZ ILs crossed the skin membrane faster than DPZ free base.

### 3.4. Acrylic Adhesive and DPZ IL Compatibility Test

A single layer or multilayer drug-in-adhesive patch is the most common type of TDDS and is mainly composed of adhesives. Acrylic adhesive and DPZ IL compatibility test was conducted to determine if there was any interaction between DPZ ILs and acrylic adhesive. DURO-TAK^®^ 87-2074, DURO-TAK^®^ 87-2051, and DURO-TAK^®^ 87-2196, common acrylic adhesives used in the manufacture of patches, were chosen as model acrylic adhesive for further investigation. These pressure sensitive adhesives not only help patch adhesion to the skin surface, but also control drug release [18]. Table 3 shows the properties of DURO-TAK^®^ 87-2074, DURO-TAK^®^ 87-2051, and DURO-TAK^®^ 87-2196.

DPZ ILs are water soluble, whereas acrylic pressure sensitive adhesives have hydrophobic properties. This may cause problems in the mixing. In this study, DPZ IL patch solution was prepared by a completely transparent state while mixing DPZ IL with acrylic adhesive agents to produce a patch without mixing problem and skin irritation effect. Figure 7 shows patch solutions prepared using DURO-TAK^®^ 87-2074, DURO-TAK^®^ 87-2051, and DURO-TAK^®^ 87-2196.

The ILs that formed transparent solutions with pressure sensitive adhesives, could be prepared easily by mixing and possible skin irritation is minimized. DURO-TAK^®^ 87-2051 showed transparent and clear appearance when incorporated with DPZ ILs containing either adipic acid, pimelic acid, suberic acid, azelaic acid, or sebacic acid. DURO-TAK^®^ 87-2074 showed transparent and clear appearance when incorporated with DPZ ILs containing either glutaric acid, itaconic acid, phthalic acid, adipic acid, pimelic acid, suberic acid, azelaic acid, or sebacic acid. DURO-TAK^®^ 87-2196 showed transparent and clear appearance when incorporated with DPZ ILs containing either glutaric acid, adipic acid, pimelic acid, suberic acid, azelaic acid, or sebacic acid.

### 3.5. DPZ IL Patch Formulation Preparation

Adipic acid and azelaic acid were chosen as model co-formers. DPZ-adipic acid and DPZ-azelaic acid were used as model ILs. Drug-in-adhesive patches were prepared by solvent evaporation. DPZ-adipic acid and DPZ-azelaic acid molar ratios were 1:1, respectively. The specific composition of different patches (DPZ, adhesive and other ingredients) is listed in Table 4, Table 5 and Table 6. DPZ ILs were dissolved in ethanol, mixed with DURO-TAK^®^ adhesive, PEG 400, Tween 80 and stirred until the mixture became homogeneous.

In this study, our DPZ transdermal patch possesses a drug-in-adhesive layer to apply on the skin and release drug in place. The release mechanism of the drug from this type of patch is diffusion across the skin. The release mechanism of the drug from this type of patch is diffusion across the skin. Valia et al. reported two types of TDDS of DPZ: (1) a drug reservoir-in-adhesive and (2) a drug matrix-in-adhesive. Drug reservoir-in-adhesive patch includes a liquid DPZ compartment separated from the adhesive layers by a membrane, and the rate of release follows a zero order. To dissolve DPZ into a liquid form in the drug reservoir system, permeation enhancers and gelling agents are required. Rate-controlling membranes were also employed to hold the liquid DPZ compartment together. The disadvantages of this type of TDDS are high cost and high potential of skin irritation. Drug matrix-in-adhesive has an adhesive layer comprising a matrix type adhesive. The drug-in-adhesive embodiment can reduce the lag time during which DPZ travels from the drug reservoir into and through the adhesive layer, thus resulting in a faster influx of DPZ into the bloodstream [19].

Transforming DPZ into DPZ ILs to promote its skin permeability is an ideal approach for the development of TDDS of DPZ. DPZ ILs were formed by pairing with several fatty acids, docusate, and ibuprofen by Hao Wu et al. Although the solubility of DPZ in water wasn’t significantly much enhanced in all cases of ILs, the use of IL technology can allow the increase of hydrophilicity and/or lipophilicity of choice [8].

In addition, PEG 400 and Tween 80 were respectively used as plasticizers and surfactants to improve the permeability of DPZ ILs. Due to the structurally organized proteins and lipids barrier properties of the skin being an obstacle to the movement of drugs under the skin, permeation enhancers (fatty acids) are generally employed to aid the transport of drugs to ensure therapeutically relevant doses reach the systemic circulation. Tween-80 is a polyethylene sorbitol ester, comprised of oleic acid as the primary fatty acid. Tween 80 was reported to be the best among all penetration enhancers used in the formulation and evaluation of transdermal patches of donepezil [20].

### 3.6. Ex Vivo Skin Permeability Test

To evaluate the permeability of the prepared patches, an ex vivo skin permeability test was performed using 6-week-old, female, hairless SKH1 mouse models. Figure 8, Figure 9 and Figure 10 show the results of ex vivo skin permeability test. The permeability of DPZ IL patches was compared with that of DPZ (non-co-former) patch. DURO-TAK^®^ 87-2051 DPZ IL patches showed the best skin permeability among the three pressure sensitive adhesives. DPZ-azelaic acid IL patch showed higher permeability compared to DPZ-adipic acid IL patch. Addition of Tween 80 and/or PED 400 increased the patch permeability. In addition, addition of PEG 400 in the patch formulations increased the skin permeability to a greater extent more than that of Tween 80.

Compatibility and optimization studies were performed to optimize the DPZ IL patch formulation. DURO-TAK^®^ 87-2051, DURO-TAK^®^ 87-2074 and DURO-TAK^®^ 87-2196 were selected, formulated, and compared. DURO-TAK^®^ 87-2051 DPZ IL patches show the best skin permeability among three pressure sensitive adhesives. This can be explained by the fact that DURO-TAK^®^ 87-2051 being the only adhesive without the presence of a cross-linker, therefore its interaction with the DPZ-ILs was minimized. DURO-TAK^®^ 87-2051 exhibits the highest tack, solid content among the three studied adhesives. Since a higher tack indicates faster bond between two surfaces, the acrylate DURO-TAK^®^ 87-2051 drug-in-adhesive may contain a higher amount of drug than that of other two adhesives. In contrast, DURO-TAK^®^ 87-2074 with two functional groups exhibited poor permeability, which may be due to the possible interaction between DURO-TAK^®^ 87-2074 and dicarboxylic based ILs.

ILs could be directly applied to the skin without further dissolving in organic solvents and/or gelling agents. Both DPZ-adipic acid IL and DPZ-azelaic acid IL show better in vitro skin permeation when they were applied directly on Strat-M™ artificial skin membrane. The cumulative permeation amount of DPZ-adipic acid IL and DPZ-azelaic acid IL were 65 times and 30 times more the cumulative permeation amount of DPZ-adipic acid IL and DPZ-azelaic acid IL loaded DURO-TAK^®^ in ex vivo, respectively. The first reason is the differences between the ex vivo and in vitro absorption ratio and the skin models used. This also indicated that even though DPZ dicarboxylic acid IL possesses very good solubility in water compared to raw DPZ, being a hydrophilic agent means the process of dissolving and diffusing in cellular bound water molecules whereas lipophilic agents may pass rapidly across the stratum corneum through the lipid-rich intercellular space. Another attribute is, in the patch, the adhesive agent works as a matrix carrier for the drug, the release of DPZ from this matrix was slowed down, reduced the time in which patient attains suitable blood-plasma levels.

The adverse events with oral donepezil (5 mg per oral, once a day initially, may increase to 10 mg every day after 4–6 weeks; may further increase to 23 mg/day after 3 months if warranted) are cholinergic hyper-stimulation symptoms. These symptoms are dose-related and largely depend on the plasma fluctuations. When donepezil is orally administered, it is well absorbed and reaches peak plasma concentrations within 4 h then decreases until the next dose [21]. In this study, a single patch containing a donepezil dose of 400 mg was maintained for 24 h and the amount of donepezil moving through or into the tissue/membrane was continuously increased. A transdermal patch would maintain drug exposure up to weeks meanwhile prolong the time to reach peak plasma concentration, reduce peak plasma concentration and keep a steady state of donepezil plasma concentration.

## 4. Conclusions

The present study is the first, to the best of our knowledge, to describe room temperature ionic liquids (ILs) of donepezil (DPZ) and carboxylic acids. Thirteen DPZ ILs were successfully prepared, which was confirmed by thermal analysis, FT-IR analysis, and NMR analysis. DPZ ILs exhibited better solubility in water and PBS buffer than that of DPZ free base. Adipic acid and azelaic acid were chosen as model co-formers. After compatibility test, DPZ IL patches were formulated using DURO-TAK^®^ 87-2051, DURO-TAK^®^ 87-2074 and DURO-TAK^®^ 87-2196. The novel DPZ IL patches inhibited crystallization and showed high skin permeability. The DPZ-azelaic ionic liquid patch formulation containing PEG 400 exhibited the best permeability. However, further studies on drug pharmacokinetic, toxicity, cytotoxicity, cell viability, and blood–brain barrier parallel permeability studies for this novel transdermal system are yet to be done. The DPZ IL transdermal patch is expected to improve patients’ compliance and product stability.

## 5. Patent

This work has been filed as Korean Registration Patent No. 10-1770675; Korean Registration Patent No. 10-1553207.

## Figures and Tables

**Figure 1 pharmaceutics-14-00205-f001:**
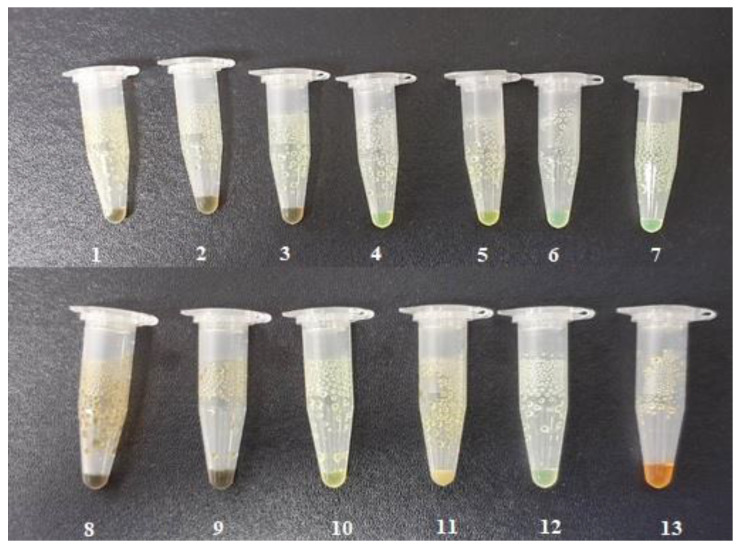
Appearances of the formed ILs: (1) DPZ-adipic acid; (2) DPZ-azelaic acid; (3) DPZ-glutaric acid; (4) DPZ-itaconic acid; (5) DPZ-maleic acid; (6) DPZ-malic acid; (7) DPZ-phthalic acid; (8) DPZ-pimelic acid; (9) DPZ-sebacic acid; (10) DPZ-succinic acid; (11) DPZ-suberic acid; (12) DPZ-tartaric acid and (13) DPZ-α ketoglutaric acid.

**Figure 2 pharmaceutics-14-00205-f002:**
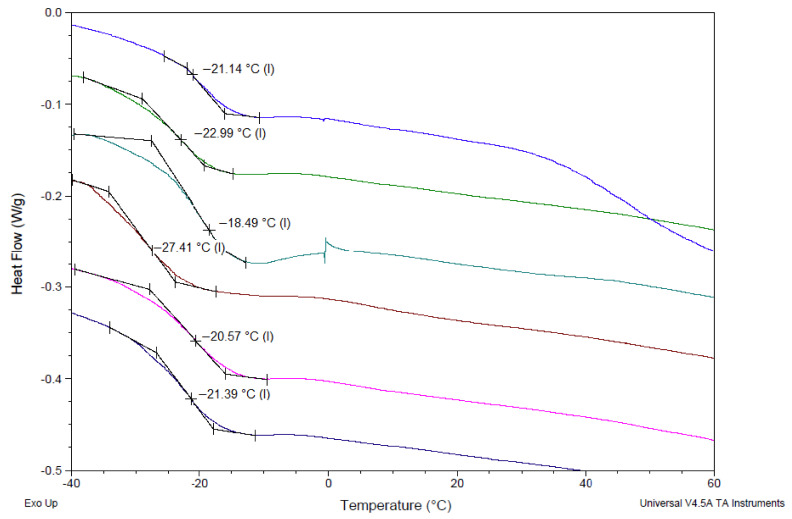
DSC thermograms of the DPZ ILs. The Tg values of DPZ-adipic acid, DPZ-azelaic acid, DPZ-glutaric acid, DPZ-itaconic acid, DPZ-maleic acid, DPZ-malic acid, DPZ-phthalic acid, DPZ-pimelic acid, DPZ-sebacic acid, DPZ-succinic acid, DPZ-suberic acid, DPZ-tartaric acid and DPZ-α ketoglutaric acid were −21.14 °C, −22.99 °C, −18.49 °C, −27.41 °C, −20.57 °C, −21.39 °C, −11 °C, −29.36 °C, −15.19 °C, −21.74 °C, −17.52 °C, −13.83 °C, and −13.10 °C.

**Figure 3 pharmaceutics-14-00205-f003:**
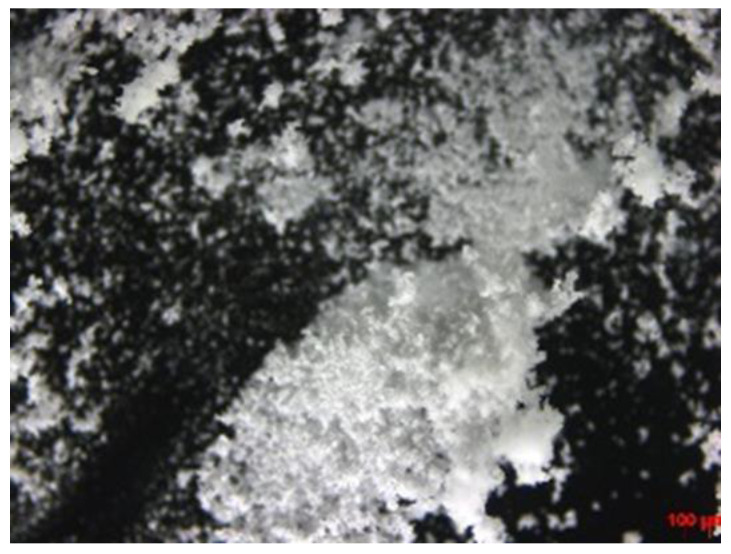
DPZ white powder under optical microscopy, scale bar: 100 μm.

**Figure 4 pharmaceutics-14-00205-f004:**
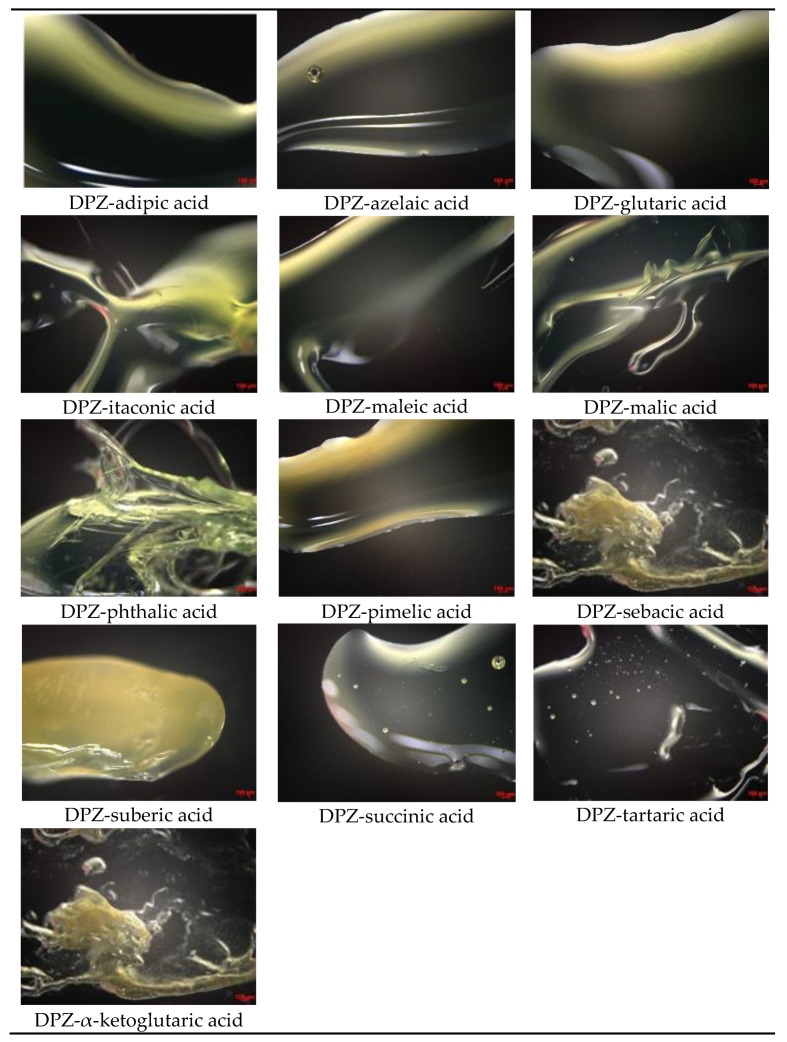
Optical microscopic images of DPZ ILs.

**Figure 5 pharmaceutics-14-00205-f005:**
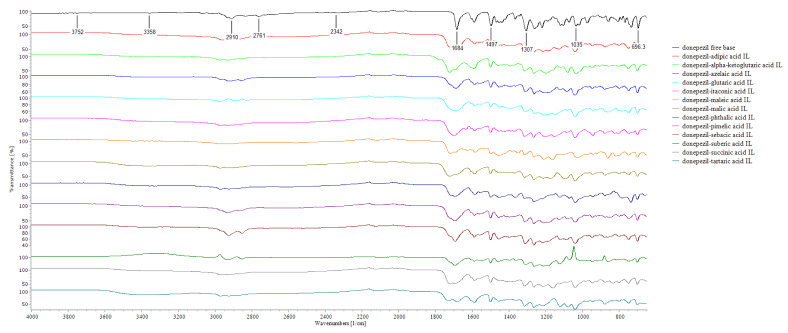
FT-IR spectra of DPZ and DPZ ILs.

**Figure 6 pharmaceutics-14-00205-f006:**
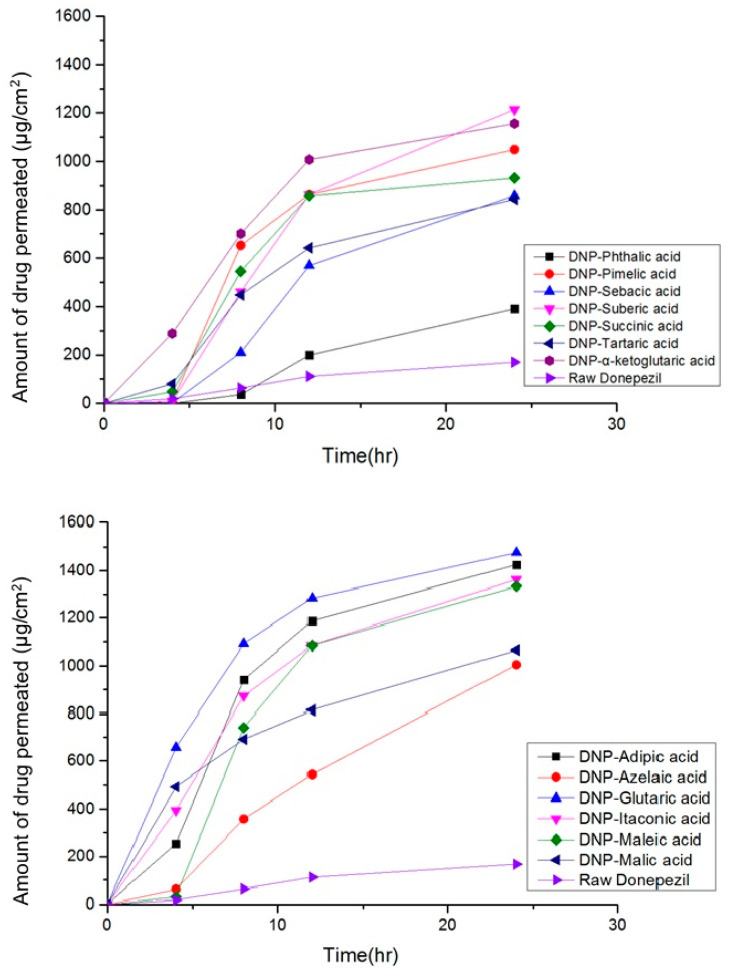
Results of skin permeation test of donepezil and various conformers ionic liquids prepared using different co-formers and donepezil (mean, *n* = 3).

**Figure 7 pharmaceutics-14-00205-f007:**
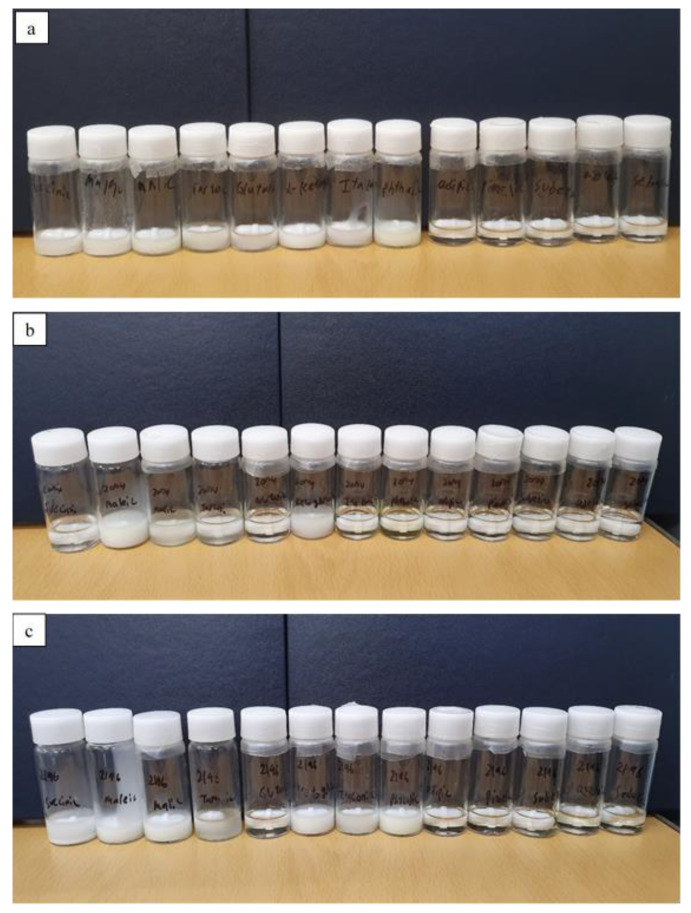
DPZ-IL patch solution: DPZ-succinic acid, DPZ-maleic acid, DPZ-malic acid, DPZ-tartaric acid, DPZ-glutaric acid, DPZ-α-ketoglutaric acid, DPZ-itaconic acid, DPZ-phthalic acid, DPZ-adipic acid, DPZ-pimelic acid, DPZ-suberic acid, DPZ-azelaic acid, DPZ-sebacic acid ILs with (**a**) DURO-TAK^®^ 87-2051, (**b**) DURO-TAK^®^ 87-2074, (**c**) DURO-TAK^®^ 87-2196.

**Figure 8 pharmaceutics-14-00205-f008:**
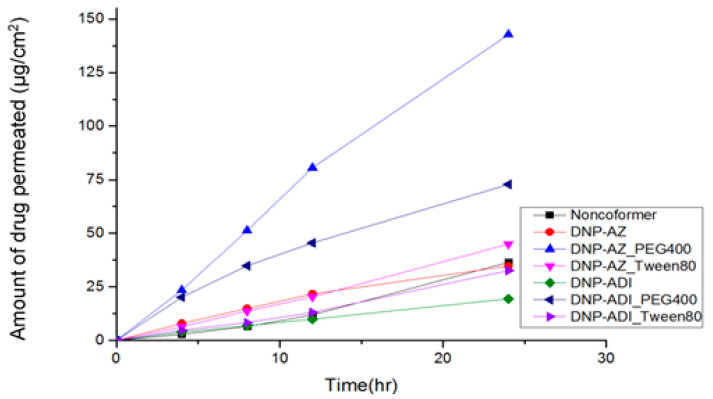
Ex vivo skin permeability of DURO-TAK^®^ 87-2051 DPZ IL patch formulations (mean, *n* = 3).

**Figure 9 pharmaceutics-14-00205-f009:**
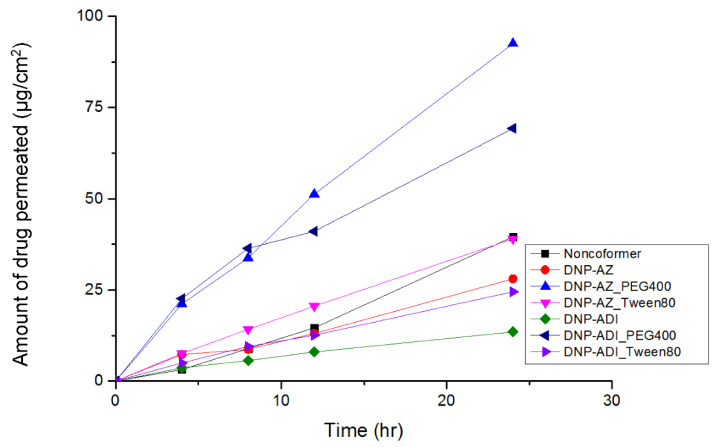
Ex vivo skin permeability of DURO-TAK^®^ 87-2074 DPZ IL patch formulation. (mean, *n* = 3).

**Figure 10 pharmaceutics-14-00205-f010:**
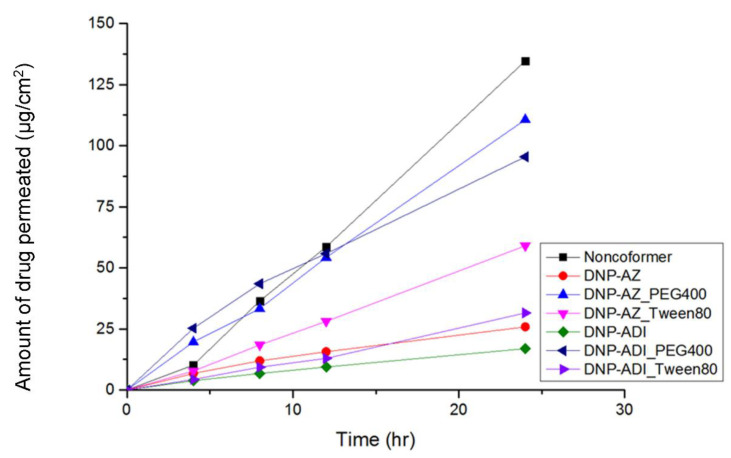
Ex vivo skin permeability of DURO-TAK^®^ 87-2196 DPZ IL patch formulation. (mean, *n* = 3).

**Table 1 pharmaceutics-14-00205-t001:** List of co-former candidates.

List of Dicarboxylic Acid Co-Former Candidates		
adipic acid	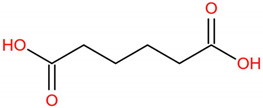	146.14 g/mol
aspartic acid	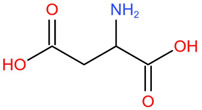	133.11 g/mol
azelaic acid	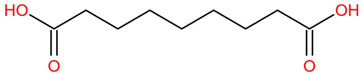	188.22 g/mol
fumaric acid	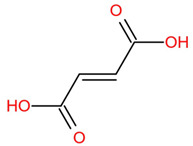	116.07 g/mol
glutamic acid	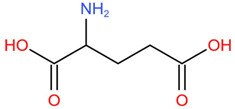	147.13 g/mol
glutaric acid	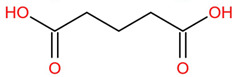	132.12 g/mol
isophthalic acid	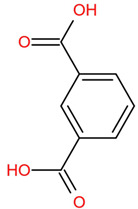	166.14 g/mol
itaconic acid	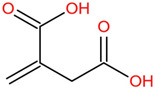	130.099 g/mol
maleic acid	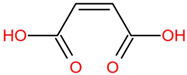	116.1 g/mol
malic acid	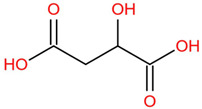	134.0874 g/mol
phthalic acid	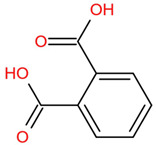	166.14 g/mol
pimelic acid	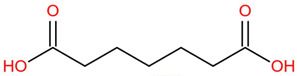	160.17 g/mol
saccharic acid	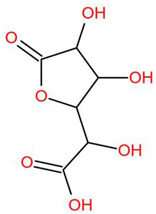	210.1388 g/mol
sebacic acid	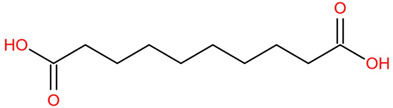	202.25 g/mol
suberic acid	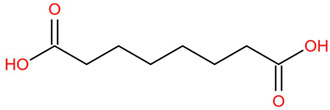	174.2 g/mol
succinic acid	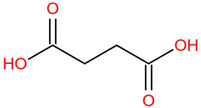	118.09 g/mol
terephthalic acid	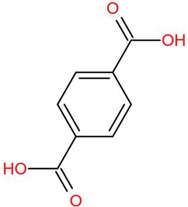	166.13 g/mol
tartaric acid	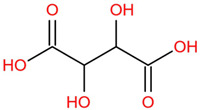	150.087 g/mol

**Table 2 pharmaceutics-14-00205-t002:** Solubility of DPZ free base and DPZ ILs in water and 0.01 M PBS pH 7.4. mean ± S.D (*n* = 3).

Compound	Solubility in Water (mg/mL)	Solubility in 0.01 M PBS pH 7.4 (mg/mL)
DPZ free base	0.007 ± 0.002	0.018 ± 0.001
DPZ—adipic acid IL	1.554 ± 0.235	1.561 ± 0.134
DPZ—azelaic acid IL	0.264 ± 0.046	1.308 ± 0.234
DPZ—glutaric acid IL	0.977 ± 0.105	1.250 ± 0.024
DPZ—itaconic acid IL	1.015 ± 0.028	1.915 ± 0.164
DPZ—maleic acid IL	0.677 ± 0.147	1.316 ± 0.129
DPZ—malic acid IL	1.892 ± 0.092	2.116 ± 0.055
DPZ—phthalic acid IL	0.313 ± 0.141	0.628 ± 0.106
DPZ—pimelic acid IL	0.660 ± 0.134	1.405 ± 0.227
DPZ—sebacic acid IL	0.287 ± 0.122	0.916 ± 0.058
DPZ—suberic acid IL	0.519 ± 0.136	1.686 ± 0.060
DPZ—succinic acid IL	1.283 ± 0.055	1.892 ± 0.243
DPZ—tartaric acid IL	1.407 ± 0.400	1.521 ± 0.067
DPZ—alpha ketoglutaric IL	1.894 ± 0.043	1.908 ± 0.226

**Table 3 pharmaceutics-14-00205-t003:** Properties of DURO-TAK^®^ 87-2074, DURO-TAK^®^ 87-2051, and DURO-TAK^®^ 87-219.

DURO-TAK^®^	Chemical Composition	Functional Groups	Cross-Linker	Solvent Composition	Tack	Solid Content	Viscosity
					(oz/in^2^)	(%)	(mPa.s)
87-2074	acrylate	–COOH–OH	present	2-propanol, ethyl acetate, toluene	20	29.5	1500
87-2051	acrylate-vinyl acetate	–COOH	not present	ethyl acetate, N-heptane, vinyl acetate, cyclohexane	80	51.5	4000
87-2196	acrylate-vinyl acetate	–COOH	present	2-propanol, ethyl acetate, toluene, vinyl acetate	20	45	2100

**Table 4 pharmaceutics-14-00205-t004:** DPZ IL patch formulation using DURO-TAK^®^ 87-2051.

DPZ	DURO-TAK^®^ 87-2051	Adipic Acid	Azelaic Acid	PEG 400	Tween 80	Ethanol
g (%)	g (%)	g (%)	g (%)	g (%)	g (%)	(mL)
0.4 (20)	2.8 (72.5)	0.15 (7.5)				0.5
0.4 (20)	2.68 (62.5)	0.15 (7.5)		0.2 (10)		0.5
0.4 (20)	2.796 (72)	0.15 (7.5)			0.01 (0.5)	0.5
0.4 (20)	2.8 (72.5)		0.15 (7.5)			0.5
0.4 (20)	2.68 (62.5)		0.15 (7.5)	0.2 (10)		0.5
0.4 (20)	2.796 (72)		0.15 (7.5)		0.01 (0.5)	0.5

**Table 5 pharmaceutics-14-00205-t005:** DPZ IL patch formulation using DURO-TAK^®^ 87-2074.

DPZ	DURO-TAK^®^ 87-2074	Adipic Acid	Azelaic Acid	PEG 400	Tween 80	Ethanol
g (%)	g (%)	g (%)	g (%)	g (%)	g (%)	(mL)
0.4 (20)	4.89 (72.5)	0.15 (7.5)				0.5
0.4 (20)	4.68 (62.5)	0.15 (7.5)		0.2 (10)		0.5
0.4 (20)	4.88 (72)	0.15 (7.5)			0.01 (0.5)	0.5
0.4 (20)	4.89 (72.5)		0.15 (7.5)			0.5
0.4 (20)	4.68 (62.5)		0.15 (7.5)	0.2 (10)		0.5
0.4 (20)	4.89 (72)		0.15 (7.5)		0.01 (0.5)	0.5

**Table 6 pharmaceutics-14-00205-t006:** DPZ IL patch formulation using DURO-TAK^®^ 87-2196.

DPZ	DURO-TAK^®^ 87-2196	Adipic Acid	Azelaic Acid	PEG 400	Tween 80	Ethanol
g (%)	g (%)	g (%)	g (%)	g (%)	g (%)	(mL)
0.4 (20)	3.21 (72.5)	0.15 (7.5)				0.5
0.4 (20)	3.07 (62.5)	0.15 (7.5)		0.2 (10)		0.5
0.4 (20)	3.2 (72)	0.15 (7.5)			0.01 (0.5)	0.5
0.4 (20)	3.21 (72.5)		0.15 (7.5)			0.5
0.4 (20)	3.07 (62.5)		0.15 (7.5)	0.2 (10)		0.5
0.4 (20)	3.2 (72)		0.15 (7.5)		0.01 (0.5)	0.5

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
