# Peer review of "Formulation, Preparation, Characterization, and Evaluation of Dicarboxylic Ionic Liquid Donepezil Transdermal Patches"

_pharmaceutics, 2022, doi:10.3390/pharmaceutics14010205_

Round 1

Reviewer 1 Report

Dear Editor,

The manuscript entitled “Formulation, preparation, characterization and evaluation of dicarboxylic ionic liquid donepezil transdermal patches” described preparation of various formulations of anti-Alzheimer drug donepezil (DPZ) and dicarboxylic acids (co-former), their attempted characterization as ionic liquid (DPZ-ILs) and posterior studies and applications of these formulations in DPZ delivery from transdermal patch containing some other additives too. Donepezil is a promising drug in Alzheimer treatment but can cause gastrointestinal side effects. Therefore, a DPZ transdermal delivery is considered an alternative for drug delivery, but traditional DPZ transdermal preparations contain many permeation enhancers owing to its low skin permeability, which may cause skin irritation problem and moreover solid crystals are generated during storage, resulting in reduce adhesion and a non-uniform skin permeation rate. The authors managed to prepare interesting and stable formulations and characterize them by NMR, FTIR and DSC. They managed to increase very significantly DPZ solubility in water and PBS as well as permeability (proven in in vitro studies) and manage to prepare patch formulations (particularly so-called DPZ-azelaic ionic liquid patch formulation containing PEG 400) that exhibited the best permeability in ex vivo studies. Hence, the work is interesting, innovative and promising (although not yet fully explored– eg. Missing skin irritation tests) and should be published but authors need to put in consideration some major corrections.

1st comment: In table 1 authors mentioned candidates of dicarboxylic acid (co-formers) they tested for formulations. Later on, they omit some of them - supposing they did not manage to prepare stable formulations. This is not a surprise as some of these dicarboxilic diacids are not strong acids e.g.  amminoacids or have some other problems (eg. Terephtalic acid). The donepezil is itself a weak base (pKa value 9.0 by pubchem) so it is still difficult to know if prepared formulations are fully ionized in order to be considered protic ionic liquids with equimolar proportion of donepezil and dicarboxylic acids or just equimolar mixtures. How does author know that? It seems that prepared mixtures are stable but maybe donepezil and dicarboxylic acid form even more stable deep eutectic mixtures in some other non-equimolar proportions. The obligatory presence of water (line 305-307) seems to appoint to this. Considering still lack of consensus for DES/protic ILs delineation among academics I believe that in order to prove their standpoint the authors should at list cite more literature about similar protic ionic liquid systems – particularly containing drugs.

2nd comment: Yet another important comment about proportion of DPZ and co-formers in order to be considered protic ionic liquids: the best way to prove equimolar proportion is by 1H-NMR. However, I find this part of the characterizations the weakest part of the manuscript. The spectra in the manuscript are completely unreadable (figure 5). Maybe the authors here have limited space, but the same problem appears in supporting info too. Signals in NMR spectra and their integrations should be clearly visible, and one spectrum should occupy the whole page in SI.  Here is another weak point of the manuscript. What kind of impurities the authors mention in their NMR spectra (line 298) and where do they come from? In my opinion it seems that authors used unwittingly deuterated chloroform for characterization that can react with DPZ (tertiary amine)-see for example Hansen S. H. and Nordholm, N. Journal of Chromatography, 204 (1981) 97-101. The authors should consider repeating NMR characterization if unable to obtain spectra without impurities maybe in D2O + TSP ideally.

3rd comment: Finally, the solubility and permeability studies show nice results but are slightly inflated by unit used. In case of ionic liquid containing active pharmaceutical ingredients, we should never forget the contribution of counterion to total molar mass (particularly if the counterion is heavy). The best way to avoid that is to use molar instead of mass concentration units for comparation. By changing the scale, author will obtain slightly diminished solubility/permeability results, but they would clearly indicate the real increase in DPZ solubility/permeability.

Thank you for your attention and collaboration.

Author Response

Dear Reviewer, 

Enclosed please find the revised manuscript entitled “Formulation, preparation, characterization and evaluation of dicarboxylic ionic liquid donepezil transdermal patches” for the consideration as an article in Pharmaceutics.

We are grateful for the thoughtful comments of the reviewers, whose contribution to the clarity and accuracy of the manuscript is substantial. Below is a point-by-point description of how we have modified the manuscript according to the reviewer’s comments or otherwise answered reviewer’s questions. On almost all points, we have been able to modify the manuscript exactly as suggested by the reviewers.

Our responses are highlighted in Bold & blue in this letter, and the corresponding revisions in the body of the manuscript are highlighted in red.

Yours sincerely,

Sung-Joo Hwang

Firstly, thank you so much for your valuable comments. All revisions in the manuscript are revised accordingly.

Line 9: Tel: +82 32 7494518.

Line 364: Figure 5

Line 401: Table 5, 6 and 7

Line 474: This work was supported by the Mid-Career Researcher Program and Basic Research Infrastructure Support Program (University-Centered Labs-2018R1A6A1A03023718) through the National Research Foundation of Korea (NRF) funded by the Korean government (MSIT)

Line 479: approval date: May 2018.

Reviewer 1:

The manuscript entitled “Formulation, preparation, characterization and evaluation of dicarboxylic ionic liquid donepezil transdermal patches” described preparation of various formulations of anti-Alzheimer drug donepezil (DPZ) and dicarboxylic acids (co-former), their attempted characterization as ionic liquid (DPZ-ILs) and posterior studies and applications of these formulations in DPZ delivery from transdermal patch containing some other additives too. Donepezil is a promising drug in Alzheimer treatment but can cause gastrointestinal side effects. Therefore, a DPZ transdermal delivery is considered an alternative for drug delivery, but traditional DPZ transdermal preparations contain many permeation enhancers owing to its low skin permeability, which may cause skin irritation problem and moreover solid crystals are generated during storage, resulting in reduce adhesion and a non-uniform skin permeation rate. The authors managed to prepare interesting and stable formulations and characterize them by NMR, FTIR and DSC. They managed to increase very significantly DPZ solubility in water and PBS as well as permeability (proven in in vitro studies) and manage to prepare patch formulations (particularly so-called DPZ-azelaic ionic liquid patch formulation containing PEG 400) that exhibited the best permeability in ex vivo studies. Hence, the work is interesting, innovative and promising (although not yet fully explored– eg. Missing skin irritation tests) and should be published but authors need to put in consideration some major corrections.

  1. In table 1 authors mentioned candidates of dicarboxylic acid (co-formers) they tested for formulations. Later on, they omit some of them - supposing they did not manage to prepare stable formulations. This is not a surprise as some of these dicarboxilic diacids are not strong acids e.g. amminoacids or have some other problems (eg. terephtalic acid). The donepezil is itself a weak base (pKa value 9.0 by pubchem) so it is still difficult to know if prepared formulations are fully ionized in order to be considered protic ionic liquids with equimolar proportion of donepezil and dicarboxylic acids or just equimolar mixtures. How does author know that? It seems that prepared mixtures are stable but maybe donepezil and dicarboxylic acid form even more stable deep eutectic mixtures in some other non-equimolar proportions. The obligatory presence of water (line 305-307) seems to appoint to this. Considering still lack of consensus for DES/protic ILs delineation among academics I believe that in order to prove their standpoint the authors should at list cite more literature about similar protic ionic liquid systems – particularly containing drugs.

(Response)

The field of study “ionic liquid” is very broad and complex. As we were aware of introducing a    number of “new DPZ-dicarboxylic acid ILs” whose properties and phase behavior were not fully investigated in this study.

The making of DPZ-coformer mixtures are mentioned section 2.2.1 and 2.2.2, the process was simply mixing DPZ and co-former (1: 1 weight ratio) and the confirmation of ionic liquids in this study was first and foremost based on visual appearance and thermal properties.

Line 107: A number of dicarboxylic acids (co-former candidates) were selected for the preliminary study. The synthesis of ILs consists in an acid-base reaction. The base, in this case DPZ free base, was mixed with the acid, respectively).

Line 111: The solvent was then evaporated at 60 °C overnight in a dry oven (Lab companion, Daejeon, Republic of Korea). After the solvent was completely evaporated, visual observation and characterization analysis of the samples was performed. The mixture was then put in a dry oven (Lab companion, Daejeon, Republic of Korea) at 60 °C overnight to evaporate the solvent and impurities (residual non-reacted and excess water). After the evaporation process, visual observation and characterization analysis of the samples was performed. The obtained ILs presented as a limpid and viscous liquid.

Line 243: DPZ free base reacted with the acid and the reaction products are a salt of DPZ-coformer complex, the protic ionic liquid itself [15].

[15] Alvarez V.H., Dosil N., Gonzáles-Cabaleiro R., Mattedi S., Martin-Pastor M., Iglesias M., Navaza J.M., 2010, Brønsted ionic liquids for sustainable process: synthesis and physical properties. Journal of Chemical & Engineering Data, 55, 625-632

Line 246: In the case of not forming ILs, the solid form was maintained at room temperature. This can be explained by the complexity of the ions forming ILs, the formation of ILs may require more than the number cations and anions present in the prepared mixture and a certain degree of ionization of acid or base ions needed should be calculated. So far, a common trait which is used to determine an IL is melting point, nevertheless, the scarce knowledge of the built-in ions of ILs is still a barrier to completely characterize an IL [12]. Aspartic acid, fumaric acid, glutamic acid, isophthalic acid, saccharic acid and terephthalic acid did not form ILs with DPZ, respectively. The appearances of the formed ILs is summarized in Figure 1.

(Line 305 – 307): According to the authors, the obligatory presence of water seen in NMR and FTIR results are in agreement with Valderrama et al., the water here is belong/entrapped inside the IL system, make it “liquid-y” not the excess water that already evaporated during the synthesis.

  1. Yet another important comment about proportion of DPZ and co-formers in order to be considered protic ionic liquids: the best way to prove equimolar proportion is by 1H-NMR. However, I find this part of the characterizations the weakest part of the manuscript. The spectra in the manuscript are completely unreadable (figure 5). Maybe the authors here have limited space, but the same problem appears in supporting info too. Signals in NMR spectra and their integrations should be clearly visible, and one spectrum should occupy the whole page in SI. Here is another weak point of the manuscript. What kind of impurities the authors mention in their NMR spectra (line 298) and where do they come from? In my opinion it seems that authors used unwittingly deuterated chloroform for characterization that can react with DPZ (tertiary amine)-see for example Hansen S. H. and Nordholm, N. Journal of Chromatography, 204 (1981) 97-101. The authors should consider repeating NMR characterization if unable to obtain spectra without impurities maybe in D2O + TSP ideally.

(Response)

We understand our weakness in performing our NMR experiment and interpretation of NMR data. For applications involving mixture of compounds, acidic and/or moisture-sensitive (very soluble) compounds, we didn’t purify deuterated chloroform prior to use (Deuterated Solvents for NMR - Cambridge Isotope Laboratories’ instructions).

We checked our CDCl3 solvent again with the same NMR equipment and the standard CDCL3 peak is attached in SI. In the standard solvent spectra, there are 3 peaks (Chloroform-D at 7.25, H20 at 1.55 and TMS base at 0).

Thank you so much for pointing out the possible reaction between DPZ (tertiary amine) and chloroform. As we noted that the chloroform will react with amines, the measurements were taken right after the NMR samples preparation to minimize possible reaction. We took in consideration this information and figured that even though there were impurities/possible reaction, molar ratio of 1:1 of donepezil to anion can still be confirmed.

Line 147: Taken in consideration that the chloroform can react with the amines [14], NMR were taken as soon as possible to minimize possible reaction.

[14] Hansen, S.H.; Nordholm, L. N-alkylation of tertiary aliphatic amines by chloroform, dichloromethane and 1,2-dichloroethane. Journal of Chromatography A 1981, 204, 97-101, doi:https://doi.org/10.1016/S0021-9673(00)81643-X.

Figure 5 was removed and all the NMR data are organized and magnified (one spectrum per page) in Supporting Information.

The impurities we were considered were: the possible reaction products of chloroform and amines, the excess water and/or the water present in the ILs system and the peak of water showed in the standard solvent NMR, and the negative chemical shifts in ppm our results (which we believe is due to the shielding effect of the aromatic ring current on proton donation).

Line 310:  A molar ratio of 1:1 between DPZ and conformer was calculated by 1H NMR for all ILs. The detailed NMR spectrum of each IL can be found in Supporting Materials section. Impurities were omitted for clearer presentation.

Line 309: This is in agreement with Hao Wu et al.’s results where the 1H NMR spectra of DPZ-docusate (a two-fold carboxylic ester), ibuprofen (a carboxylic acid) and several fatty acids were obtained at various concentrations in deuterated dimethyl sulfoxide (DMSO) [8]

One more thing that we’d like to address is using D2O + TSP as solvent like your suggestion.

We really want to perform the NMR again with the more ideal solvent as such, but there are some obstacles that we’re having now: 1) The solvent is not available now at our facility, it comes with a high price and also take longer time to deliver (as we have to submit our response within a short time, again, please carefully consider if repeating NMR characterization using D20 +TSP is a must for us). 2) If our result is in agreement with other paper in which other type of solvent was used and the possible reaction between CDCl3 was avoid by technique, is repeating NMR a must?)

  1. Finally, the solubility and permeability studies show nice results but are slightly inflated by unit used. In case of ionic liquid containing active pharmaceutical ingredients, we should never forget the contribution of counterion to total molar mass (particularly if the counterion is heavy). The best way to avoid that is to use molar instead of mass concentration units for comparation. By changing the scale, author will obtain slightly diminished solubility/permeability results, but they would clearly indicate the real increase in DPZ solubility/permeability.

(Response)

Thanks so much for this comment.

The paper is written based on the work filed as Korean Registration Patent No. 10-1770675; Korean Registration Pa-tent No. 10-1553207, in the patent the unit used was mass instead of molar.

We would like to add this opinion in our discussion as below:

Line 46: DPZ, 2-[(1-benzyl-4-piperidyl) methyl]-5,6-dimethoxy-2,3-dihydroinden-1-one, 179.31 g/mol (free base).

Table 1 is updated with molar mass (g/mol) information.

Line 349: In case of ILs containing active pharmaceutical ingredients, the contribution of couterions to total molar mass should be taken in consideration therefore mass per volume unit can be converted to mol per volume unit for better comparison.

Reviewer 2:

This paper described a drug-in-adhesive patch with donepezil. The novelty is related to ionic liquid technology, promoting solubility and skin permeability of donepezil free-base. Promising results of this work, reported by the authors, must be encouraged.

  1. Were sink conditions assured in the (section 2.2.8.) in vitro skin permeation test?

(Response)

Yes, sink conditions were maintained throughout the Franz cell experiment.

We considered using large volume receptor chamber (12 mL), the large volume reduces the gradient, so buildup of the compound in the receptor part is not a problem (sink conditions are maintained).

Line 160: The Franz diffusion cell system consists of a 2 mL donor compartment and a 12 mL receptor compartment.

Line 168: At predetermined time points, 1 mL of solution was taken from the receiver compartment and immediately replaced with 1 mL of the fresh preheated medium. This was to maintain a constant volume as well as sink conditions. Air bubbles underneath the membrane were removed.

  1. What criteria were applied to select Tween 80 as an enhancer to promote skin permeability of donepezil?

(Response)

Line 407: Due to the structurally organized proteins and lipids barrier properties of the skin being an obstacle to the movement of drugs under the skin, permeation enhancers (fatty acids) are generally employed to aid the transport of drugs to ensure therapeutically relevant doses reach the systemic circulation. Tween-80 is a polyethylene sorbitol ester, comprised of oleic acid as the primary fatty acid.  Tween 80 (0.83 % w/w) was reported to be the best among all penetration enhancers used in the formulation and evaluation of transdermal patches of donepezil by Madan JR et al [19].

[19]      Madan, J.R.; Argade, N.S.; Dua, K. Formulation and evaluation of transdermal patches of donepezil. Recent patents on drug delivery & formulation 2015, 9, 95-103, doi:10.2174/1872211308666141028213615.

  1. Figures 5 and 6: I suggest to include the error bars

(Response)

Thanks for your suggestion.

In response to Reviewer 1, we decided to remove Figures 5 and present all the NMR data/spectra in Supporting Materials.

  1. Tables 6, 7, and 8: I suggest to place them in material and methods

(Response)

Thanks for your suggestion.

The specific composition of different patches (DPZ, adhesive and other ingredients) is listed in Tables 7, 8, and 9. Here in, the amount of DPZ and coformer (Adipic acid/Azelaic acid) (g) were calculated based on a molar ratio 1:1 between DPZ and conformer which was found in 1H NMR spectra. Therefore, we’d like to mention patch composition after 1H NMR results.

  1. Figures 8 and 9: The title of Y-axis is lacking. And the units.

(Response)

Figures 7, 8, 9: The title and unit “Amount of drug permeated (μg/cm2)” of Y-axis was added.

  1. I suggest discussing whether the amount of donepezil that permeates across the skin is equivalent to the oral therapeutic dose.

(Response)

Line 443: The adverse events with oral donepezil (5 mg per oral, qHS initially, may increase to 10 mg qDay after 4-6 weeks; may further increase to 23 mg/day after 3 months if warranted) are cholinergic hyperstimulation symptoms. These symptoms are dose-related and largely depend on the plasma fluctuations. When donepezil is orally administered, it is well absorbed and reaches peak plasma concentrations within 4 h then decreases until the next dose [18]. In this study, a single patch containing a donepezil dose of 400 mg was maintained for 24 h and the amount of donepezil moving through or into the tissue/membrane was continuous increased. A transdermal patch would maintain drug exposure up to weeks meanwhile prolong the time to reach peak plasma concentration, reduce peak plasma concentration and keep a steady state of donepezil plasma concentration.

[20]      Kim, Y.H.; Choi, H.Y.; Lim, H.S.; Lee, S.H.; Jeon, H.S.; Hong, D.; Kim, S.S.; Choi, Y.K.; Bae, K.S. Single dose pharmacokinetics of the novel transdermal donepezil patch in healthy volunteers. Drug design, development and therapy 2015, 9, 1419-1426, doi:10.2147/dddt.S78555

  1. There are other papers exploring strategies to administer donepezil through the skin. I recommend comparing the results of this work with those from previously published papers.

(Response)

Thank you for your comments.

The result of this work is compared with that of Hao Wu [12] where donepezil-ILs were formed with other type of coformers.

In this study, not only 13 new ILs were found but also the ILs show better solubility and skin permeation, other than that, a formulation was developed with an excellent performance and compatibility with PEG.

Reviewer 2 Report

This paper described a drug-in-adhesive patch with donepezil. The novelty is related to ionic liquid technology, promoting solubility and skin permeability of donepezil free-base. Promising results of this work, reported by the authors, must be encouraged.

Some considerations:

1) Were sink conditions assured in the (section 2.2.8.) in vitro skin permeation test?

2) What criteria were applied to select Tween 80 as an enhancer to promote skin permeability of donepezil?

3) Figures 5 and 6: I suggest to include the  error bars

4) Tables 6, 7, and 8: I suggest to place them in material and methods

5) Figures 8 and 9: The title of Y-axis is lacking. And the units.

6) I suggest discussing whether the amount of donepezil that permeates across the skin is equivalent to the oral therapeutic dose.

7) There are other papers exploring strategies to administer donepezil through the skin. I recommend comparing the results of this work with those from previously published papers.

Author Response

Author's Reply to the Review Report

-------------------------------------------------------------------

Yonsei University

College of Pharmacy

Songdogwahak-ro, Yeonsu-gu, Incheon 406-840

Republic of Korea

Tel (82)32-749-4518 

Fax(82)32-749-4105

Email: sjh11@yonsei.ac.kr

Enclosed please find the revised manuscript entitled “Formulation, preparation, characterization and evaluation of dicarboxylic ionic liquid donepezil transdermal patches” for the consideration as an article in Pharmaceutics.

We are grateful for the thoughtful comments of the reviewers, whose contribution to the clarity and accuracy of the manuscript is substantial. Below is a point-by-point description of how we have

modified the manuscript according to the reviewer’s comments or otherwise answered reviewer’s questions. On almost all points, we have been able to modify the manuscript exactly as suggested by the reviewers.

Our responses are highlighted in Bold & blue in this letter, and the corresponding revisions in the body of the manuscript are highlighted in red.

Yours sincerely,

Sung-Joo Hwang

General Comments:

Firstly, thank you so much for your valuable comments. All revisions in the manuscript are revised accordingly.

Line 9: Tel: +82 32 7494518.

Line 364: Figure 5

Line 401: Table 5, 6 and 7

Line 474: This work was supported by the Mid-Career Researcher Program and Basic Research Infrastructure Support Program (University-Centered Labs-2018R1A6A1A03023718) through the National Research Foundation of Korea (NRF) funded by the Korean government (MSIT)

Line 479: approval date: May 2018.

Reviewer 1:

The manuscript entitled “Formulation, preparation, characterization and evaluation of dicarboxylic ionic liquid donepezil transdermal patches” described preparation of various formulations of anti-Alzheimer drug donepezil (DPZ) and dicarboxylic acids (co-former), their attempted characterization as ionic liquid (DPZ-ILs) and posterior studies and applications of these formulations in DPZ delivery from transdermal patch containing some other additives too. Donepezil is a promising drug in Alzheimer treatment but can cause gastrointestinal side effects. Therefore, a DPZ transdermal delivery is considered an alternative for drug delivery, but traditional DPZ transdermal preparations contain many permeation enhancers owing to its low skin permeability, which may cause skin irritation problem and moreover solid crystals are generated during storage, resulting in reduce adhesion and a non-uniform skin permeation rate. The authors managed to prepare interesting and stable formulations and characterize them by NMR, FTIR and DSC. They managed to increase very significantly DPZ solubility in water and PBS as well as permeability (proven in in vitro studies) and manage to prepare patch formulations (particularly so-called DPZ-azelaic ionic liquid patch formulation containing PEG 400) that exhibited the best permeability in ex vivo studies. Hence, the work is interesting, innovative and promising (although not yet fully explored– eg. Missing skin irritation tests) and should be published but authors need to put in consideration some major corrections.

  1. In table 1 authors mentioned candidates of dicarboxylic acid (co-formers) they tested for formulations. Later on, they omit some of them - supposing they did not manage to prepare stable formulations. This is not a surprise as some of these dicarboxilic diacids are not strong acids e.g. amminoacids or have some other problems (eg. terephtalic acid). The donepezil is itself a weak base (pKa value 9.0 by pubchem) so it is still difficult to know if prepared formulations are fully ionized in order to be considered protic ionic liquids with equimolar proportion of donepezil and dicarboxylic acids or just equimolar mixtures. How does author know that? It seems that prepared mixtures are stable but maybe donepezil and dicarboxylic acid form even more stable deep eutectic mixtures in some other non-equimolar proportions. The obligatory presence of water (line 305-307) seems to appoint to this. Considering still lack of consensus for DES/protic ILs delineation among academics I believe that in order to prove their standpoint the authors should at list cite more literature about similar protic ionic liquid systems – particularly containing drugs.

(Response)

The field of study “ionic liquid” is very broad and complex. As we were aware of introducing a    number of “new DPZ-dicarboxylic acid ILs” whose properties and phase behavior were not fully investigated in this study.

The making of DPZ-coformer mixtures are mentioned section 2.2.1 and 2.2.2, the process was simply mixing DPZ and co-former (1: 1 weight ratio) and the confirmation of ionic liquids in this study was first and foremost based on visual appearance and thermal properties.

Line 107: A number of dicarboxylic acids (co-former candidates) were selected for the preliminary study. The synthesis of ILs consists in an acid-base reaction. The base, in this case DPZ free base, was mixed with the acid, respectively).

Line 111: The solvent was then evaporated at 60 °C overnight in a dry oven (Lab companion, Daejeon, Republic of Korea). After the solvent was completely evaporated, visual observation and characterization analysis of the samples was performed. The mixture was then put in a dry oven (Lab companion, Daejeon, Republic of Korea) at 60 °C overnight to evaporate the solvent and impurities (residual non-reacted and excess water). After the evaporation process, visual observation and characterization analysis of the samples was performed. The obtained ILs presented as a limpid and viscous liquid.

Line 243: DPZ free base reacted with the acid and the reaction products are a salt of DPZ-coformer complex, the protic ionic liquid itself [15].

[15] Alvarez V.H., Dosil N., Gonzáles-Cabaleiro R., Mattedi S., Martin-Pastor M., Iglesias M., Navaza J.M., 2010, Brønsted ionic liquids for sustainable process: synthesis and physical properties. Journal of Chemical & Engineering Data, 55, 625-632

Line 246: In the case of not forming ILs, the solid form was maintained at room temperature. This can be explained by the complexity of the ions forming ILs, the formation of ILs may require more than the number cations and anions present in the prepared mixture and a certain degree of ionization of acid or base ions needed should be calculated. So far, a common trait which is used to determine an IL is melting point, nevertheless, the scarce knowledge of the built-in ions of ILs is still a barrier to completely characterize an IL [12]. Aspartic acid, fumaric acid, glutamic acid, isophthalic acid, saccharic acid and terephthalic acid did not form ILs with DPZ, respectively. The appearances of the formed ILs is summarized in Figure 1.

(Line 305 – 307): According to the authors, the obligatory presence of water seen in NMR and FTIR results are in agreement with Valderrama et al., the water here is belong/entrapped inside the IL system, make it “liquid-y” not the excess water that already evaporated during the synthesis.

  1. Yet another important comment about proportion of DPZ and co-formers in order to be considered protic ionic liquids: the best way to prove equimolar proportion is by 1H-NMR. However, I find this part of the characterizations the weakest part of the manuscript. The spectra in the manuscript are completely unreadable (figure 5). Maybe the authors here have limited space, but the same problem appears in supporting info too. Signals in NMR spectra and their integrations should be clearly visible, and one spectrum should occupy the whole page in SI. Here is another weak point of the manuscript. What kind of impurities the authors mention in their NMR spectra (line 298) and where do they come from? In my opinion it seems that authors used unwittingly deuterated chloroform for characterization that can react with DPZ (tertiary amine)-see for example Hansen S. H. and Nordholm, N. Journal of Chromatography, 204 (1981) 97-101. The authors should consider repeating NMR characterization if unable to obtain spectra without impurities maybe in D2O + TSP ideally.

(Response)

We understand our weakness in performing our NMR experiment and interpretation of NMR data. For applications involving mixture of compounds, acidic and/or moisture-sensitive (very soluble) compounds, we didn’t purify deuterated chloroform prior to use (Deuterated Solvents for NMR - Cambridge Isotope Laboratories’ instructions).

We checked our CDCl3 solvent again with the same NMR equipment and the standard CDCL3 peak is attached in SI. In the standard solvent spectra, there are 3 peaks (Chloroform-D at 7.25, H20 at 1.55 and TMS base at 0).

Thank you so much for pointing out the possible reaction between DPZ (tertiary amine) and chloroform. As we noted that the chloroform will react with amines, the measurements were taken right after the NMR samples preparation to minimize possible reaction. We took in consideration this information and figured that even though there were impurities/possible reaction, molar ratio of 1:1 of donepezil to anion can still be confirmed.

Line 147: Taken in consideration that the chloroform can react with the amines [14], NMR were taken as soon as possible to minimize possible reaction.

[14] Hansen, S.H.; Nordholm, L. N-alkylation of tertiary aliphatic amines by chloroform, dichloromethane and 1,2-dichloroethane. Journal of Chromatography A 1981, 204, 97-101, doi:https://doi.org/10.1016/S0021-9673(00)81643-X.

Figure 5 was removed and all the NMR data are organized and magnified (one spectrum per page) in Supporting Information.

The impurities we were considered were: the possible reaction products of chloroform and amines, the excess water and/or the water present in the ILs system and the peak of water showed in the standard solvent NMR, and the negative chemical shifts in ppm our results (which we believe is due to the shielding effect of the aromatic ring current on proton donation).

Line 310:  A molar ratio of 1:1 between DPZ and conformer was calculated by 1H NMR for all ILs. The detailed NMR spectrum of each IL can be found in Supporting Materials section. Impurities were omitted for clearer presentation.

Line 309: This is in agreement with Hao Wu et al.’s results where the 1H NMR spectra of DPZ-docusate (a two-fold carboxylic ester), ibuprofen (a carboxylic acid) and several fatty acids were obtained at various concentrations in deuterated dimethyl sulfoxide (DMSO) [8]

One more thing that we’d like to address is using D2O + TSP as solvent like your suggestion.

We really want to perform the NMR again with the more ideal solvent as such, but there are some obstacles that we’re having now: 1) The solvent is not available now at our facility, it comes with a high price and also take longer time to deliver (as we have to submit our response within a short time, again, please carefully consider if repeating NMR characterization using D20 +TSP is a must for us). 2) If our result is in agreement with other paper in which other type of solvent was used and the possible reaction between CDCl3 was avoid by technique, is repeating NMR a must?)

  1. Finally, the solubility and permeability studies show nice results but are slightly inflated by unit used. In case of ionic liquid containing active pharmaceutical ingredients, we should never forget the contribution of counterion to total molar mass (particularly if the counterion is heavy). The best way to avoid that is to use molar instead of mass concentration units for comparation. By changing the scale, author will obtain slightly diminished solubility/permeability results, but they would clearly indicate the real increase in DPZ solubility/permeability.

(Response)

Thanks so much for this comment.

The paper is written based on the work filed as Korean Registration Patent No. 10-1770675; Korean Registration Pa-tent No. 10-1553207, in the patent the unit used was mass instead of molar.

We would like to add this opinion in our discussion as below:

Line 46: DPZ, 2-[(1-benzyl-4-piperidyl) methyl]-5,6-dimethoxy-2,3-dihydroinden-1-one, 179.31 g/mol (free base).

Table 1 is updated with molar mass (g/mol) information.

Line 349: In case of ILs containing active pharmaceutical ingredients, the contribution of couterions to total molar mass should be taken in consideration therefore mass per volume unit can be converted to mol per volume unit for better comparison.

Reviewer 2:

This paper described a drug-in-adhesive patch with donepezil. The novelty is related to ionic liquid technology, promoting solubility and skin permeability of donepezil free-base. Promising results of this work, reported by the authors, must be encouraged.

  1. Were sink conditions assured in the (section 2.2.8.) in vitro skin permeation test?

(Response)

Yes, sink conditions were maintained throughout the Franz cell experiment.

We considered using large volume receptor chamber (12 mL), the large volume reduces the gradient, so buildup of the compound in the receptor part is not a problem (sink conditions are maintained).

Line 160: The Franz diffusion cell system consists of a 2 mL donor compartment and a 12 mL receptor compartment.

Line 168: At predetermined time points, 1 mL of solution was taken from the receiver compartment and immediately replaced with 1 mL of the fresh preheated medium. This was to maintain a constant volume as well as sink conditions. Air bubbles underneath the membrane were removed.

  1. What criteria were applied to select Tween 80 as an enhancer to promote skin permeability of donepezil?

(Response)

Line 407: Due to the structurally organized proteins and lipids barrier properties of the skin being an obstacle to the movement of drugs under the skin, permeation enhancers (fatty acids) are generally employed to aid the transport of drugs to ensure therapeutically relevant doses reach the systemic circulation. Tween-80 is a polyethylene sorbitol ester, comprised of oleic acid as the primary fatty acid.  Tween 80 (0.83 % w/w) was reported to be the best among all penetration enhancers used in the formulation and evaluation of transdermal patches of donepezil by Madan JR et al [19].

[19]      Madan, J.R.; Argade, N.S.; Dua, K. Formulation and evaluation of transdermal patches of donepezil. Recent patents on drug delivery & formulation 2015, 9, 95-103, doi:10.2174/1872211308666141028213615.

  1. Figures 5 and 6: I suggest to include the error bars

(Response)

Thanks for your suggestion.

In response to Reviewer 1, we decided to remove Figures 5 and present all the NMR data/spectra in Supporting Materials.

  1. Tables 6, 7, and 8: I suggest to place them in material and methods

(Response)

Thanks for your suggestion.

The specific composition of different patches (DPZ, adhesive and other ingredients) is listed in Tables 7, 8, and 9. Here in, the amount of DPZ and coformer (Adipic acid/Azelaic acid) (g) were calculated based on a molar ratio 1:1 between DPZ and conformer which was found in 1H NMR spectra. Therefore, we’d like to mention patch composition after 1H NMR results.

  1. Figures 8 and 9: The title of Y-axis is lacking. And the units.

(Response)

Figures 7, 8, 9: The title and unit “Amount of drug permeated (μg/cm2)” of Y-axis was added.

  1. I suggest discussing whether the amount of donepezil that permeates across the skin is equivalent to the oral therapeutic dose.

(Response)

Line 443: The adverse events with oral donepezil (5 mg per oral, qHS initially, may increase to 10 mg qDay after 4-6 weeks; may further increase to 23 mg/day after 3 months if warranted) are cholinergic hyperstimulation symptoms. These symptoms are dose-related and largely depend on the plasma fluctuations. When donepezil is orally administered, it is well absorbed and reaches peak plasma concentrations within 4 h then decreases until the next dose [18]. In this study, a single patch containing a donepezil dose of 400 mg was maintained for 24 h and the amount of donepezil moving through or into the tissue/membrane was continuous increased. A transdermal patch would maintain drug exposure up to weeks meanwhile prolong the time to reach peak plasma concentration, reduce peak plasma concentration and keep a steady state of donepezil plasma concentration.

[20]      Kim, Y.H.; Choi, H.Y.; Lim, H.S.; Lee, S.H.; Jeon, H.S.; Hong, D.; Kim, S.S.; Choi, Y.K.; Bae, K.S. Single dose pharmacokinetics of the novel transdermal donepezil patch in healthy volunteers. Drug design, development and therapy 2015, 9, 1419-1426, doi:10.2147/dddt.S78555

  1. There are other papers exploring strategies to administer donepezil through the skin. I recommend comparing the results of this work with those from previously published papers.

(Response)

Thank you for your comments.

The result of this work is compared with that of Hao Wu [12] where donepezil-ILs were formed with other type of coformers.

In this study, not only 13 new ILs were found but also the ILs show better solubility and skin permeation, other than that, a formulation was developed with an excellent performance and compatibility with PEG.

Round 2

Reviewer 1 Report

Dear Editor,

There are significant improvements in the manuscript except for the 1H-NMR.  The spectra are now visible in SI and the problem of deuterated solvent chosen (CDCl3) is obvious: many phantom picks, so-called impurities referred by authors seems to appear not due to actual impurities present but due to insufficient dissolution or ion pairing and associations of ions and other species that could be present when this partly ionic formulation is dissolved in nonpolar CDCl3. This makes NMR interpretation - signal integration- difficult and I believe the problem could be solved if more polar deuterated solvent was used (D2O, MeOD). On the other side since the authors cannot wait to buy another solvent and repeat characterization, the formulations are simple equimolar mixtures which otherwise characterized seems to be uniform and some basic information of the proportion is visible in NMR I will not further insist on improvement. 

Author Response

Dear Reviewer, 

We're grateful for your thoughtful comments. On almost all points, we've been trying to modify our manuscript as suggested. 

Reviewer 2 Report

The answers by the authors were evasive and unclear.

Sink conditions were achieved? What is the solubility of the donepezil free-base in the medium? And how sink conditions were defined?

There are a lot of papers reporting donepezil transdermal, but the authors did not report them. I strongly recommend including and discussing these papers. The major point to achieve success in the transdermal administration of donepezil is to achieve a significant permeation (transdermal flux) across the skin, without skin toxicity. The comparison of transdermal flux among different papers would help to better understand this challenge and to demonstrate if this paper really represents an innovation.

Author Response

Dear Reviewer, 

We're grateful for your thoughtful comments. On almost all points, we've been able to modify our manuscript as suggested. 

Round 3

Reviewer 2 Report

The authors improved the manuscript overall quality.